



# When and where does near-surface runoff occur in a pre-Alpine headwater catchment?

Victor A. Gauthier[1], Anna Leuteritz[1], Ilja van Meerveld[1]

[1]Department of Geography, Zurich University, Zurich, Switzerland

*Correspondence to*: Victor A. Gauthier (victor.gauthier@lilo.org)

**Abstract.** Although runoff processes have been described for many locations worldwide, there has been a lack of studies for poorly drained soils where most of the runoff may occur near the soil surface. Therefore, in this study, we aimed to improve the understanding of near-surface processes across a small headwater catchment with low permeable gleysols, which is typical for the Swiss pre-Alpine environment. We installed 14 small (1 m x 3 m) 10 bounded runoff plots to collect overland flow (including biomat flow; OF) and shallow subsurface flow through the permeable topsoil, which we refer to as topsoil interflow (TIF). The runoff plots were located at different topographic locations and had a range of vegetation covers. For 27 rainfall events during the summer of 2022, we determined the occurrence and amount of OF and TIF. OF and TIF occurred for approximately half of the events, but the frequency of occurrence depended on the topographic wetness index (TWI) and vegetation cover. The 15 runoff ratios (ratio between runoff produced divided by the total precipitation) increased with increasing precipitation and antecedent wetness conditions but did not correlate with the maximum rainfall intensity. Runoff ratios were highly variable and were generally higher for TIF than OF. Runoff ratios for OF were larger than one for some plots, indicating the occurrence lateral inflow to the plot from outside. Runoff ratio did not change after removing the upper boundary of the plot, suggesting that the actual flow-path lengths over the surface are short. 20 Overall, this study highlights the importance of fast near-surface processes in pre-Alpine catchments underlain by low permeability gleysols, and that these processes occur across a range of catchment locations and land covers.


## 1 Introduction

Lateral flow from hillslopes is an important contributor to streamflow during rainfall and snowmelt events, and
can transport considerable amounts of nutrients, solutes and sediment to the stream network. However, hillslope
runoff processes are highly variable (e.g., Bachmair and Weiler, 2012) and nonlinear (e.g., Penna et al., 2011;
Tromp-van Meerveld and McDonnell, 2006; Vreugdenhil et al., 2022), which means that not all hillslopes
contribute equally to streamflow, nor contribute during all events (Ambroise, 2016; Anderson and Burt, 1978b;
Rinderer et al., 2014; Uchida and Asano, 2010). Spatially, runoff generation depends on topography (Anderson
and Burt, 1978a; Tromp-van Meerveld and McDonnell, 2006), microtopography (Appels et al., 2011; Polyakov et
al., 2021), vegetation cover (Gerke et al., 2015; Mishra et al., 2022), and soil and bedrock characteristics (Descroix
et al., 2001; Palmer and Smith, 2013; Uchida and Asano, 2010). Temporally, runoff generation varies with rainfall
event characteristics (Tarboton, 2003; Weiler et al., 2005) and antecedent wetness conditions (Bronstert and
Bárdossy, 1999; Henninger et al., 1976) or the combination of rainfall and antecedent wetness conditions (Detty
and McGuire, 2010; Nanda and Safeeq, 2023; Penna et al., 2011; Saffarpour et al., 2016). Despite several decades
of studies on hillslope runoff processes in temperate (Betson and Marius, 1969; Dunne and Black, 1970; Minea et
al., 2019; Tanaka et al., 1988; Weiler and Naef, 2003), semi-arid (Mounirou et al., 2012; Puigdefabregas et al.,
1998) and tropical (Bonell and Gilmour, 1978; Dunne and Dietrich, 1980; Zwartendijk et al., 2020) climates, there
are still several open questions regarding the importance of hillslope runoff processes and the factors that control
it (Blöschl et al., 2019). Most hillslope runoff studies in temperate climates have focused on hillslopes with well-
drained soils, where overland flow (OF) is unlikely to occur (Barthold and Woods, 2015). Nevertheless, high
rainfall intensity sprinkling experiments on vegetated hillslopes on low permeability gleysols in Switzerland have
shown that OF is likely an important runoff pathway (e.g., Badoux et al., 2006; Scherrer et al., 2007; Weiler et al.,
1999). For example, during sprinkling experiments on two 13 m$^2$ forested plots in the Alptal, 20% of the flow
occurred in the humic A horizon and 5% as OF (Feyen et al., 1996). Sprinkling experiments in nearby catchments
suggested that OF was an even larger fraction of the precipitation (between 39 and 94% in the study by Badoux et
al. (2006)). Dye staining experiments, furthermore, showed that most of the infiltrating water remained in the
densely rooted organic-rich topsoil, and did not infiltrate into the low permeability clay below it (Schneider et al.,
2014; Weiler et al., 1998). We refer to the lateral flow through this organic rich topsoil as topsoil interflow (TIF)
to differentiate it from the lateral subsurface flow (SSF) that is generated deeper in the soil profile (e.g., at the soil-
bedrock interface). Studies in other parts of the world have, similarly, shown that OF can be important on vegetated
hillslopes (e.g., Buttle and Turcotte, 1999; Gomi et al., 2008; Kim et al., 2014; Miyata et al., 2009), or highlighted
the importance of flow through the litter or the organic-rich topsoil due to hydrophobicity at the interface of the
organic layer and mineral soil (i.e., biomat flow; Sidle et al., 2007). Other studies have highlighted the importance
of exfiltrating subsurface flow for OF (Buttle, 1994; Buttle and McDonald, 2002; Feyen et al., 1996; Lapides et
al., 2022; Tanaka, 1982). Yet, little is known about OF and TIF generation on vegetated hillslopes in temperate
climates. In part, this is because these previous studies are mainly focused on observations (or rainfall simulation)
at a few plots. Understanding the spatiotemporal variability in OF and TIF requires measurements at various
locations for a range of events. Therefore, we set up a hydrological measurement network consisting of 14 small
runoff plots (1 m x 3 m) across the 20 ha Studibach catchment in the Alptal, Switzerland. The plots represent a
range of topographic conditions and vegetation covers. We measured OF (including biomat flow) in runoff gutters





and TIF flow rates in trenches for 27 events during the 2022 snow-free season. We used these data to address the following questions:

1) How often do OF and TIF occur, and how does this depend on the plot characteristics (vegetation, slope, topographic position)?
2) Is the spatial variation in the runoff ratios for OF and TIF related to the plot characteristics (vegetation, slope, topographic position), and how does it depend on them?
3) Is there a precipitation (amount or intensity) or antecedent wetness threshold before considerable OF and TIF occur?

A better understanding of the spatial and temporal variation in OF and TIF is necessary to develop better models or regionalize streamflow predictions (Barthold and Woods, 2015) and land management (Naef et al., 2002). Because climate change will affect large rainfall events, it will also affect the occurrence of OF and TIF, and thus related streamflow and solute and sediment transport responses.

**2 Study site**

The research was conducted in the Studibach catchment, a typical pre-Alpine headwater catchment, in the Alptal valley, located ~ 40 km southeast of Zurich in Switzerland (Coordinates: 47.038° N, 8.717° E). The geology, topography, landuse and climate are typical for the Swiss pre-Alpine area. Because most areas have a restricted soil permeability (Figure S1), it is a region where we expected to find sear-surface flow pathways.

The 20 ha Studibach catchment ranges from 1270 m to 1650 m asl. in elevation and has a mean slope of 18°, varying between 0° and 69° (based on the 0.5 m DEM (Swisstopo SwissAlti3D)). The climate is humid, with a mean annual temperature of 6°C, varying from -1°C in January to 14°C in July (Schleppi et al., 1998). The mean annual precipitation is approximately 2300 mm y$^{-1}$, of which ~ 30% falls as snow (Stähli and Gustafsson, 2006). Precipitation is evenly distributed throughout the year, but the most intense rain events occur in summer (June to September), when it rains on average every second day (Fischer et al., 2017; van Meerveld et al., 2019). About 55% of the catchment is covered by open coniferous forest (Figure 1) dominated by *Picea abies L.* with an understory of *Vaccinium sp* (Hagedorn et al., 2000). Approximately 45% of the catchment (mainly in the flatter parts and depressions) is covered by grasslands and wetlands. About 10% of the catchment (the upper part) is used as a pasture in summer (Rinderer et al., 2016).

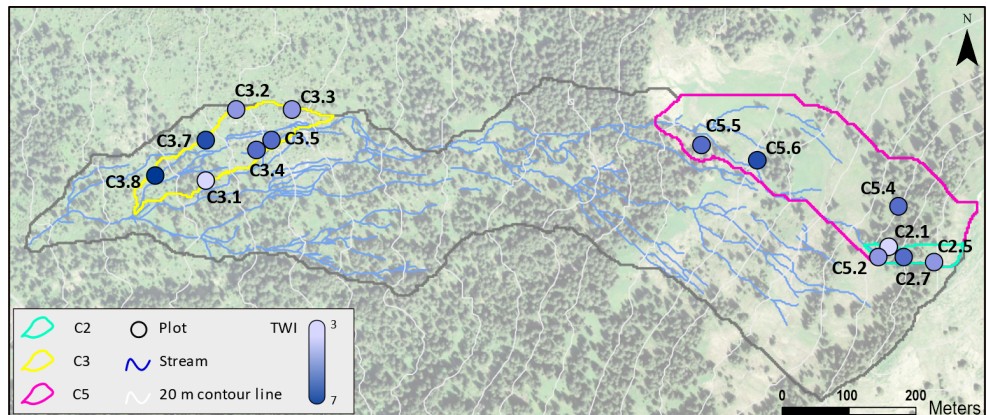

**Figure 1: Map of the Studibach catchment with the location of the plots in the three subcatchments (C2, C3 and C5), the field surveyed stream network (blue lines), and the 20 m contour lines (in gray). The background map shows the vegetation (Source: Swisstopo SwissImage (2023)). The plots are colour-coded according to the Topographic Wetness Index (darker blue colour indicates a wetter location).**

The soils are primarily silty-clay and silt-clay-loam in texture. They are underlain by low permeability, clay-rich flysch bedrock consisting of calcareous sandstone and argillite and bentonite schist layers (Mohn et al., 2000). Soil depths range from 0.5-1 m at the ridges to 2.5 m in depressions. The soil type in the steeper parts is an umbric gleysol, with an oxidized $B_w$ horizon below mor humus. In the flatter parts, where the water table is close to the surface, it is a mollic gleysol with a reduced $B_g$ horizon below a muck humus layer (Hagedorn et al., 2001; Schleppi et al., 1998).

The wet climate and low permeability soil and bedrock result in shallow groundwater levels throughout most of the catchment (Rinderer et al., 2016) and a dense drainage network (Figure 1, van Meerveld et al., 2019). The streams respond quickly to precipitation within tens of minutes. Although streamflow is dominated by pre-event water (Kiewiet et al., 2020), event water contributions can be > 50% (Fischer et al., 2017; von Freyberg et al., 2018). In a recent study, Bujak-Ozga et al. (2024) showed that the event water flux is much larger than the precipitation falling on the flowing stream network and must thus come from areas outside the flowing stream network, except at the onset of the events.

**3 Methods**

**3.1 Selection of runoff plot locations**

We installed 14 small (1 m x 3 m) bounded runoff plots in two parts of the catchment to cover the range in slope, vegetation, and wetness conditions. The selection of the locations for the plots was based on the Topographic Wetness Index (TWI; Beven and Kirkby, 1979) calculated for a 6-m resolution Digital Elevation Model (DEM). Rinderer et al. (2014) determined the distribution of TWI values for seven subcatchments, divided each distribution into eight equally sized classes, and installed a groundwater monitoring well in the pixels with the median TWI for that class. We selected three subcatchments (C2, C3 and C5; Figure 1). The C2 subcatchment in the lower Studibach has various slopes and is dominated by open coniferous forest (see Figure S2), natural clearings, and



wetlands. Subcatchment C3 is steeper and mainly forested, while C5 has moderate slopes and is mostly covered
by grasslands and wetlands.

We installed trenched runoff plots within 6 m of each well in the selected subcatchments in an area with a relatively
uniform vegetation cover and slope (Table 1), except for the well with the lowest TWI because the distance to the
ridge was too short to install a runoff plot. Because the groundwater levels and dynamics in the catchment are
strongly related to TWI (Rinderer et al., 2014; 2016), we assumed that stratification of the plots based on TWI
would result in a better representation of the variability in near-surface flow responses than a random sampling

design. Because of the stratification based on the TWI, the runoff plots differed not only in terms of topographic
position and wetness conditions but also in slope and vegetation cover (Table 1). Similar to Rinderer et al. (2014),
we refer to the plot locations as follows: 'CX.Y', where X represents the subcatchments and Y corresponds to the
TWI class, ranging from 1 (driest site) to 8 (wettest site) (Figure 1).

**3.2 Field measurements**

**3.2.1 Runoff plot construction and flow measurements**

We installed the plots in summer 2021 and collected data between May and October 2022. At each selected
location, we built a small (1 m x 3m) bounded runoff plot following the methodology of Maier and van Meerveld
(2021) and Weiler et al. (1999). At the lower end of the plot, we dug a trench until the depth of the reduced clay
layer (generally at ~ 40 cm below the soil surface; Table 1), where there are only very few visible roots. We put

drain foil on the trench face to block the lateral subsurface flow and a drainage tube at the bottom of the trench
(rolled into the foil) to collect the water and channel it via a hose to an Upwelling Bernoulli Tube (UBeTube; cf.
Stewart et al., 2015). The trench was backfilled to ensure slope stability. An OF gutter was installed on the surface.
Plastic foil was inserted down to ~ 3 cm depth to guide the runoff into a 1 m gutter. Flow from the OF gutter was
routed to another UBeTube via a hose. OF thus also includes biomat flow. A fiberglass roof covered the gutter to

avoid any direct precipitation entering into the gutter. At the sides and the upper end of the plots, we inserted
plastic lawn edging 5 cm into the ground to minimize the flow of OF into or out of the plot (see photos in Figures
S2 and S3). Note that the plastic sheeting at the top of the plot was removed on September 6[th] 2022, for another
experiment.

The UBeTubes were built from 10 cm diameter PVC pipes at the University of Zurich following the design of

Stewart et al. (2015) using a water jet cutter (see Figure S3). All UBeTubes were screened for consistency of the
V-notches before field installation.

In each UBeTube, we installed a conductivity, temperature and pressure logger (DCX-22-CTD, Keller Druck,
Switzerland). To determine the water level from these pressure measurements, we installed eight barometric
loggers (DCX-22, Keller Druck, Switzerland) throughout the catchment to measure the atmospheric pressure. Each

barometric logger was wrapped in heat-reflecting foil to minimize temperature effects (Shannon et al., 2022). All
loggers recorded the pressure at a 5-minute interval.

The water levels in the UBeTubes were converted to flow rates (Q in L min$^{-1}$) based on rating curves developed
in the laboratory for ten UBeTubes. Because the rating curves were similar for nine out of the ten UBeTubes and
the other one was visibly different (Morlang, 2022), we used the same rating curve for 26 of 28 UBeTubes: $Q =$

$\alpha h^{\beta}$, where $\alpha$ and $\beta$ are constants (respectively $0.24 \pm 0.08$ and $1.88 \pm 0.27$), and $h$ is the water level above the





bottom of the V-notch (in cm). For the two UBeTubes for which the V-notch was visibly different, we used the rating curves corresponding to their V-notch shape ($\alpha = 0.080$, $\beta = 2.269$). The flow into the UBeTubes, when the water level was below the V-notch, was based on the diameter of the UBeTubes.

### 3.1.2 Soil moisture measurements

We installed soil moisture sensors (TEROS 12 and GS3, METER Group, USA) at 5, 20 and 30 cm below the surface at the edge of six of the plots: C3.1, C3.4, C3.8, C5.2, C5.4 and C5.6. The sensors were connected to ZL6 and EM50 data loggers (METER Group, USA) that recorded the soil moisture at a 5-minute frequency.

### 3.1.2 Precipitation data

Precipitation was measured with a tipping bucket at the Erlenhöhe meteorological station located ~ 400 m from
the Studibach outlet at 1215 m asl. The data were provided by the Swiss Federal Institute for Forest, Snow and Landscape Research (WSL) and have a 10-minute resolution.

### 3.3 Plot characteristics

For each plot, we determined several characteristics (Table 1). We classified the plots according to four main vegetation types: open forest (F), natural clearings in the open forest (C), grasslands (G), and wetlands (W). Forests
are areas with large spruce trees, where the soil is covered mainly by moss or blueberry bushes (plots C2.1, C2.5, C3.1), or young trees (plot C3.5). Clearings are small open areas in the forest covered by grasses, mosses, horsetail, alpine flowers and blueberry bushes (i.e., they are natural open areas and not locations where the forest has been logged). Grasslands are large open areas dominated by grasses and alpine flowers. Wetlands are also open areas but are dominated by sphagnum moss, horsetail, alpine flowers and grasses.
The topographic wetness index (TWI) of the plots was based on the analyses of Rinderer et al. (2014). The slope of the plots was determined by measuring the difference in elevation between the top and the bottom of the plots using a self-made microtopographic profiler (cf. Leatherman, 1987).
During the trench installation, we determined the depth of the A and B horizons. In addition, soil samples were taken next to each plot at 10-15 cm below the soil surface to determine the organic matter content (OM) based on
the loss on ignition method. Many of the site characteristics are correlated with each other (see Table S3).





**Table 1: Main characteristics for the 14 plots: Topographic wetness index (TWI), soil depth at the bottom of the A and B horizons, slope, and organic matter content at 10-15 cm depth, Vegetation cover: Forest (F), Natural Clearing (C), Grassland (G), Wetland (W).**

| Location | TWI | Depth A Horizon (cm) | Depth B Horizon (cm) | Organic matter (%) | Vegetation | Slope (°) |
|----------|-----|----------------------|----------------------|--------------------|------------|-----------|
| C2.1 | 3.5 | 10 | 33 | 20 | F | 35 |
| C2.5 | 4.5 | 10 | 39 | 13 | F | 26 |
| C2.7 | 5.3 | 10 | 40 | — | C | 33 |
| C5.2 | 4.1 | 5 | 31 | 3 | G | 27 |
| C5.4 | 5.0 | 10 | 42 | 13 | G | 35 |
| C5.5 | 5.5 | 15 | 31 | 25 | W | 9 |
| C5.6 | 5.9 | 15 | > 40 | 23 | W | 14 |
| C3.1 | 3.4 | 10 | 40 | 14 | F | 13 |
| C3.2 | 4.1 | 15 | 30 | 20 | C | 19 |
| C3.3 | 4.4 | 17 | 32 | 18 | C | 18 |
| C3.4 | 4.8 | 20 | 40 | 11 | C | 15 |
| C3.5 | 5.2 | 20 | 40 | 19 | F | 27 |
| C3.7 | 6.0 | 18 | 35 | 48 | C | 21 |
| C3.8 | 7.0 | 15 | 30 | 43 | W | 11 |

### 3.4 Data analysis

#### 3.4.1 Precipitation event characteristics

We divided the measurement period into 27 events, defined as periods with more than 5 mm of precipitation, separated by at least 12 h without precipitation. For the plots in catchment C3, data were recorded for all 27 events. Measurements for the plots in subcatchments C2 and C5 started later, so data are only available for the last 20 events (E7-E27, Table S1). For each event, we determined the total precipitation ($P$), 10-min maximum precipitation intensity ($I_{10}$), mean precipitation intensity for every 30-minute period with precipitation ($I_{mean}$), and the event duration (time between the start and end of the event; $D$) (Table S1). We, furthermore, divided the events into three categories based on the mean intensity: low (< 2 mm h$^{-1}$), medium (2-4 mm h$^{-1}$) and high (> 4 mm h$^{-1}$). Not surprisingly, many of these event characteristics were correlated with each other (see Table S2). To characterize the antecedent wetness conditions for each event, we determined the Antecedent Soil-moisture Index ($ASI$; Haga et al., 2005) for the top 5 cm of the soil by multiplying the average moisture content measured at 5 cm depth at the start of the event by the 5 cm depth. To calculate the average soil moisture, we used three out of six soil moisture sensors (at C3.4, C3.8, and C5.2) that cover the range in TWI values and had the longest complete data record. We determined the $ASI$ for other depth intervals using different sensors (e.g., 0-10, 0-15, 0-25 and 0-





30 cm) as well, but these were all highly correlated ($r^2 > 0.99$). Finally, we determined the sum of $ASI$ and $P$ ($ASI+P$) for each event as a measure of the overall wetness conditions (Detty and McGuire, 2010; Penna et al., 2011).

**3.4.2 Runoff response**

For each event, we calculated the total outflow from the UBeTubes between the start of the event and 6 hours after the precipitation stopped ($Q_{OF}$ and $Q_{TIF}$), the time of the start of the response ($t_s$) (i.e., when the flow from the UBeTubes started or the flow started to increase), and the time of the peak flow rate ($t_p$). We calculated the lag times from these data by relating them to the start of the precipitation event and the peak precipitation intensity.

To compare the runoff responses for the different events, we calculated the runoff ratios for OF and TIF ($R_{OF}$ and $R_{TIF}$, respectively), by dividing the total flow ($Q_{OF}$ or $Q_{TIF}$) by the total precipitation ($P$) and the projected area of the plots. To calculate the runoff ratios, we set all total flow amounts $< 0.1$ L to zero. We took this minimum amount of flow because of the uncertainties in the water level data (i.e., it was not always clear if the event caused the water level in the UBeTubes tubes to go up by only a few mm and produce a minimal amount of flow) and because such small flow amounts are insignificant. We also determined the total amount of near-surface runoff ($Q = Q_{OF} + Q_{TIF}$) and the percentage of the near-surface flow caused by OF ($P_{OF} = Q_{OF}/Q$). We did these calculations for each event and each plot. Finally, we determined the percentage of events for which the total amount of OF or TIF was $> 0.1$ L ($F_{OF}$ and $F_{TIF}$, respectively).

**3.4.3 Statistical analyses**

To determine the influence of the event characteristics ($P$, $I_{10}$, $I_{mean}$, $D$, $ASI$, $ASI+P$) on the amount of flow ($Q_{OF}$ or $Q_{TIF}$) or the runoff ratios ($R_{OF}$ or $R_{TIF}$), we used the Spearman rank correlation ($r_s$). This was done for each plot for which there were at least four events for which flow was measured. To determine the existence of a runoff threshold, segmented regressions were conducted between the $ASI+P$ and $R_{OF}$ or $R_{TIF}$ for each plot using the 'piecewise-regression' package (Pilgrim, 2021). As there was not always an evident threshold at the derived breakpoint, we manually defined some thresholds. Similarly, we used the Spearman rank correlation ($r_s$) between the site characteristics (Table 2), the frequency of flow ($F_{OF}$ and $F_{TIF}$) during the monitoring period, and the runoff ratios ($R_{OF}$ and $R_{TIF}$) and percentage of total flow ($P_{OF}$) for the 26 out 27 events for which flow was measured for at least four, the Spearman rank correlation between the site characteristics and the runoff ratios ($R_{OF}$ and $R_{TIF}$), and OF as a percentage of total flow. As a measure of the overall relation between the site characteristics and the runoff ratios, we determined the average of the Spearman rank values across the 26 events. For the vegetation (categorical data), we used dummy variables based on the ranking (high to low) of the vegetation cover: forest (0), clearing (1), grassland (2) and wetland (3). All analyses were done in Python (version 3.12). In particular, we used the packages *Pandas*, *Scipy*, *Matplotlib* and *Seaborn*.

**4 Results**

**4.1 Occurrence of OF and TIF**

Total precipitation for the 27 events ranged between 5 and 98 mm, and the 10-min maximum intensity varied between 4.8 to 63.0 mm h$^{-1}$ (Table S1). Even though the summer of 2022 in the Alps was classified as relatively

dry (Abegg and Mayer, 2023), we measured Overland flow (OF) and topsoil interflow (TIF) for approximately half of the events (Figure 2). However, the frequency of OF and TIF ($F_{OF}$ and $F_{TIF}$, respectively) varied

considerably (Figure 2), ranging from 14 to 78% for $F_{OF}$ and between 19 and 86% for $F_{TIF}$. $F_{OF}$ and $F_{TIF}$ were for most plots similar (e.g., C3.8, C.5.5 or C5.6) or lower for OF than for TIF (e.g., C2.7, C3.1). However, there were two apparent exceptions. For the forested plots C2.1 and C2.5, OF was measured much more frequently than TIF (Figure 2).


**Figure 2: Percentage of events for which overland flow (OF; left) or topsoil interflow (TIF; right) was measured during the summer 2022 for each of the 14 plots (ordered by subcatchment and topographic wetness index (TWI)). Each bar is divided into three categories, to indicate the frequency of very small (light color), small (median color) and considerable (dark color) amounts of flow. The icons above the bars indicate the land cover. For other details about the**

**plots, see Table 1.**

**4.2 Runoff ratios**

The runoff ratios for OF and TIF ($R_{OF}$ and $R_{TIF}$, respectively) were highly variable and varied from plot to plot and event to event (Figure 3). $R_{OF}$ did not seem to be considerably affected by opening the plot border at the upper end

of the plot in September 10[th], as the ratio between the average of $R_{OF}$ before August 15[th] and the average of $R_{OF}$ after September 10[th] was 1.06 (Figure 4). The runoff ratios for OF were > 1 during events E1, E5, E10 and E24 (at up to 3 plots). Also, the plot borders were not deep enough to block lateral in or outflow for TIF and the contributing area was likely much larger than the plot for TIF. So, it is not surprising that the runoff ratios for TIF were > 1 during these events (at up to 9 plots).

**4.2.1 Temporal variation in runoff ratios**

The runoff ratios increased with increasing precipitation and antecedent wetness conditions ($ASI+P$) for many plots (Figure 3). However, this was not the case for some plots in the forest or natural clearings. For plots C2.1, C2.5 and C3.4, the runoff ratios for OF ($R_{OF}$) were considerable for events with low $ASI+P$, but did not increase in wetter conditions (Figure 3).






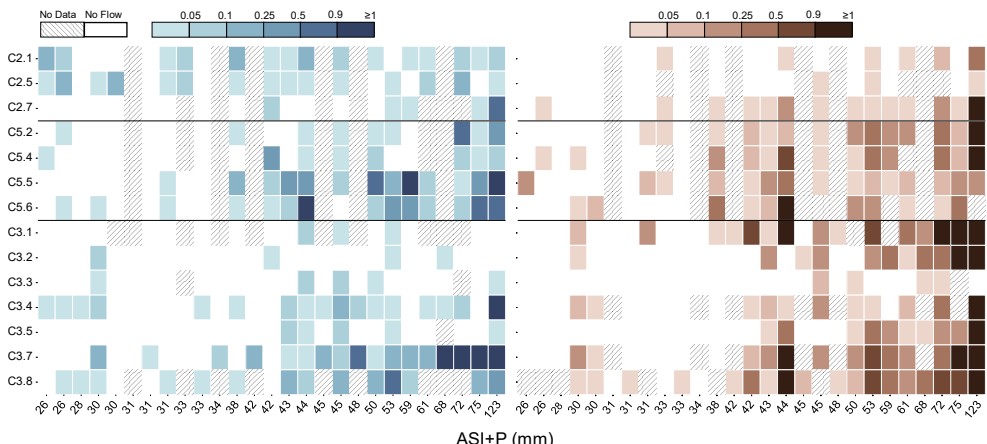

**Figure 3: Heatmap of the runoff ratio ($R$) for each event (x-axis) and each plot (y-axis) for overland flow ($R_{OF}$; left) and topsoil interflow ($R_{TIF}$; right). Events are ranked according to the $ASI+P$. For events that produced <0.1 L of flow, the runoff ratio is plotted as zero (white). All runoff ratios >1 were set to 1 for plotting. Events for which data are missing**

**are indicated with the hatched lines. See Figures S4 and S5 for the heatmaps where the events are ordered according to the total precipitation ($P$) or the mean precipitation intensity ($I_{mean}$), respectively.**

For most plots, the runoff ratios for OF and TIF were high as soon as $ASI+P$ was higher than ~ 39 mm (Figures 3 and 4). For seven of the plots, there was a clear runoff threshold for OF. For TIF, this was the case for 11 of the

14 plots (Figure 4). The Spearman rank correlation between $ASI+P$ and $R_{OF}$ varied between -0.16 and 0.83 (mean across all plots: 0.46) and was statistically significant for half of the plots. It was low ($r_s < 0.5$) and not significant for plots C2.1, C2.5, C3.1, C3.2, C3.3, C3.4 and C3.5. In general, the correlations between the runoff ratio for OF ($R_{OF}$) and $ASI+P$ were highest for the plots at higher TWI, but this was partly because more events resulted in flow for these plots (Figure 2). The Spearman rank correlation between the TWI and the correlation between the runoff

ratios and $ASI+P$ was 0.77 for OF (p < 0.01). For TIF, the Spearman rank correlation between $ASI+P$ and the runoff ratio ($R_{TIF}$) varied between 0.43 and 0.89 (mean across all plots: 0.70) and was significant for all plots (Table 2). The correlation was lowest for plots C2.1, C2.5, C3.3, and C5.5, but the strength of the relation between $ASI+P$ and $R_{TIF}$ was not related to the TWI ($r_s = 0.01$, p = 0.74).

The correlations between the runoff ratios and total precipitation were fairly similar to those with $ASI+P$ (compare

Figure 3 and S8). This was not the case for the $ASI$ alone. For plots with a low TWI, the OF ratios were negatively correlated with $ASI$, while for plots with a higher TWI, they were positively correlated with $ASI$ (Table 2; Figure S5). The Spearman rank correlation between the TWI and the correlation between the runoff ratio and $ASI$ was 0.87 for OF (p < 0.001). This relation was not observed for TIF ($r_s = 0.24$; p = 0.40).

Contrary to our expectation, no statistically significant correlation existed between the 10-min maximum rainfall

intensity ($I_{10}$) and the runoff ratio (neither for $R_{OF}$ nor $R_{TIF}$). The relation between the runoff ratio and the mean intensity was not clear either ($r_s$ ranged between 0.16 and 0.51 for OF (Table 2; Figure S5) and between -0.01 and 0.58 for TIF (Table 2; Figure S6).



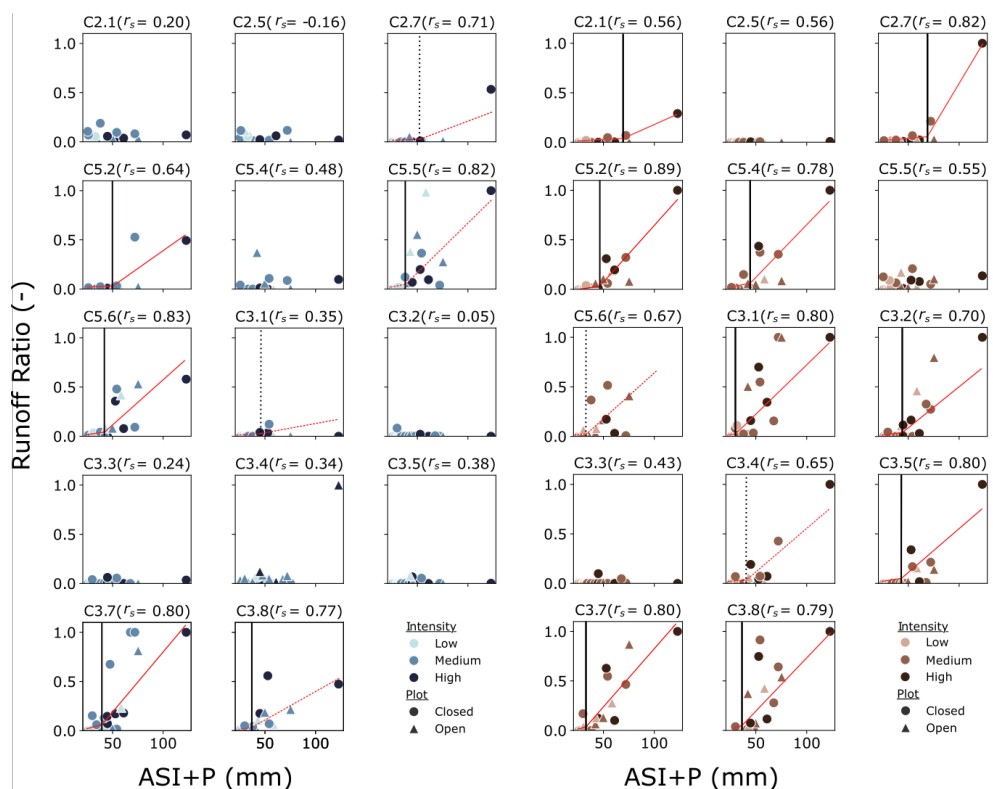

**Figure 4: Relation between the runoff ratio (R) and *ASI+P* (mm) for overland flow (R_OF; left) and topsoil interflow (R_TIF; right) for each plot. The red line indicates the results of the piecewise regression and the black line indicates the threshold (computed: solid line; determined manually: dashed line). The Spearman rank are printed above each subplot. Runoff ratios > 1 are plotted as 1 for visual clarity. Each symbol represents one event, whereby circles represent events before September 10th when the upper border was closed, and triangles represent events when the upper border was open). The colour of the symbols represents the mean intensity class: low, medium and high.**



**Table 2: The average and range (min-max) of the Spearman rank correlation between the runoff ratio and the five event characteristics for overland flow ($R_{OF}$) and topsoil interflow ($R_{TIF}$), as well as the percentage of plots for which the correlation was statistically significant at the 0.05 and 0.10 level. *P*: total precipitation (mm), $I_{10}$: 10-min maximum precipitation intensity (mm h$^{-1}$), $I_{mean}$: mean precipitation intensity for every 30-minute period with precipitation (mm h$^{-1}$), *ASI*: antecedent soil moisture index for the top 5 cm of soil (mm), *ASI+P*: antecedent soil moisture index plus total precipitation (mm).**

| | | *P* | $I_{10}$ | $I_{mean}$ | *ASI* | *ASI+P* |
|---|---|---|---|---|---|---|
| $R_{OF}$ | Average | 0.50 | 0.13 | 0.33 | 0.17 | 0.46 |
| | Range | -0.09 – 0.83 | -0.09 – 0.36 | 0.16 – 0.51 | -0.20 – 0.62 | -0.16 – 0.83 |
| | p < 0.05 | 57% | 0% | 21% | 21% | 50% |
| | p < 0.10 | 64% | 0% | 36% | 29% | 64% |
| $R_{TIF}$ | Average | 0.71 | 0.14 | 0.35 | 0.34 | 0.70 |
| | Range | 0.45 – 0.89 | -0.23 – 0.40 | -0.01 – 0.58 | 0.09 – 0.65 | 0.43 – 0.89 |
| | p < 0.05 | 100% | 0% | 36% | 29% | 100% |
| | p < 0.10 | 100% | 0% | 57% | 36% | 100% |

### 4.2.2 Spatial variation in runoff ratios

The runoff ratios of OF and TIF were positively related to the TWI and negatively correlated with the local slope, but these correlations were only statistically significant for a small fraction of the events, generally larger events with wet conditions (*ASI+P* > 39 mm; Table 3). The runoff ratios were also correlated with the vegetation cover and the organic matter content (Table 3). The correlations between $R_{OF}$ and TWI, vegetation or organic matter content were higher for events with wet conditions (Figure S6). For $R_{TIF}$, the correlations were highest at intermediate wetness conditions (*ASI+P* between 30 to 60 mm; Figure S7).





**Table 3: The average and range (min-max) of the Spearman rank correlation between the site characteristics (TWI, slope, vegetation, and organic matter content ($OM$) and the percentage of events for which OF or TIF was >0.1 L ($F_{OF}$ and $F_{TIF}$), the runoff ratios for OF and $TIF$ ($R_{OF}$ and $R_{TIF}$) or OF as a fraction of total near-surface flow ($P_{OF}$), as well as the percentage of events for which the correlations were statistically significant at the 0.05 and 0.10 level.**

|  |  | TWI | Slope | Vegetation | OM |
|---|---|---|---|---|---|
| $F_{OF}$ |  | 0.51 | -0.14 | 0.35 | 0.38 |
| $F_{TIF}$ |  | 0.51 | -0.32 | 0.64 | 0.47 |
| $R_{OF}$ | Average | 0.35 | -0.09 | 0.19 | 0.25 |
|  | Range | -0.59 – 0.80 | -0.76 – 0.59 | -0.53 – 0.76 | 0.00 – 0.88 |
|  | p< 0.05 | 19% | 4% | 11% | 12% |
|  | p< 0.10 | 27% | 12% | 22% | 19% |
| $R_{TIF}$ | Average | 0.20 | -0.34 | 0.33 | 0.17 |
|  | Range | -0.26 – 0.66 | -0.78 – 0.52 | -0.26 – 0.72 | -0.30 – 0.89 |
|  | p< 0.05 | 17% | 8% | 12% | 12% |
|  | p< 0.10 | 17% | 25% | 15% | 15% |
| $P_{OF}$ | Average | -0.02 | 0.07 | 0.12 | 0.11 |
|  | Range | -0.71– 0.57 | -0.63 – 0.71 | -0.63 – 0.70 | -0.71 – 0.57 |
|  | p< 0.05 | 6% | 6% | 12% | 0% |
|  | p< 0.10 | 6% | 6% | 12% | 0% |

### 4.3 Relative importance of OF and TIF

The fraction of total near-surface flow that flowed over the surface ($P_{OF}$) during an event varied spatially and from event to event. During dry conditions near-surface flow was dominated by OF ($P_{OF} > 0.5$). $P_{OF}$ decreased with increasing $ASI+P$ for most plots (Figure 5). The exceptions are two steep forested plots in the upper subcatchment (plots C2.5 and C2.1), and a wetland location (C5.5) that generated more OF than TIF for most events (see also Figures 2 and 3).

The relative importance of overland flow was not consistently correlated to the plot characteristics (Table 3) but depended on the event characteristics. The correlation between $P_{OF}$ and TWI increased from dry condition ($r_s = -0.71$) to wet condition ($r_s = 0.3$), while the correlation between $P_{OF}$ and slope tended to decrease from dry ($r_s = 0.71$) to wet condition ($r_s = -0.14$; Figure S8).



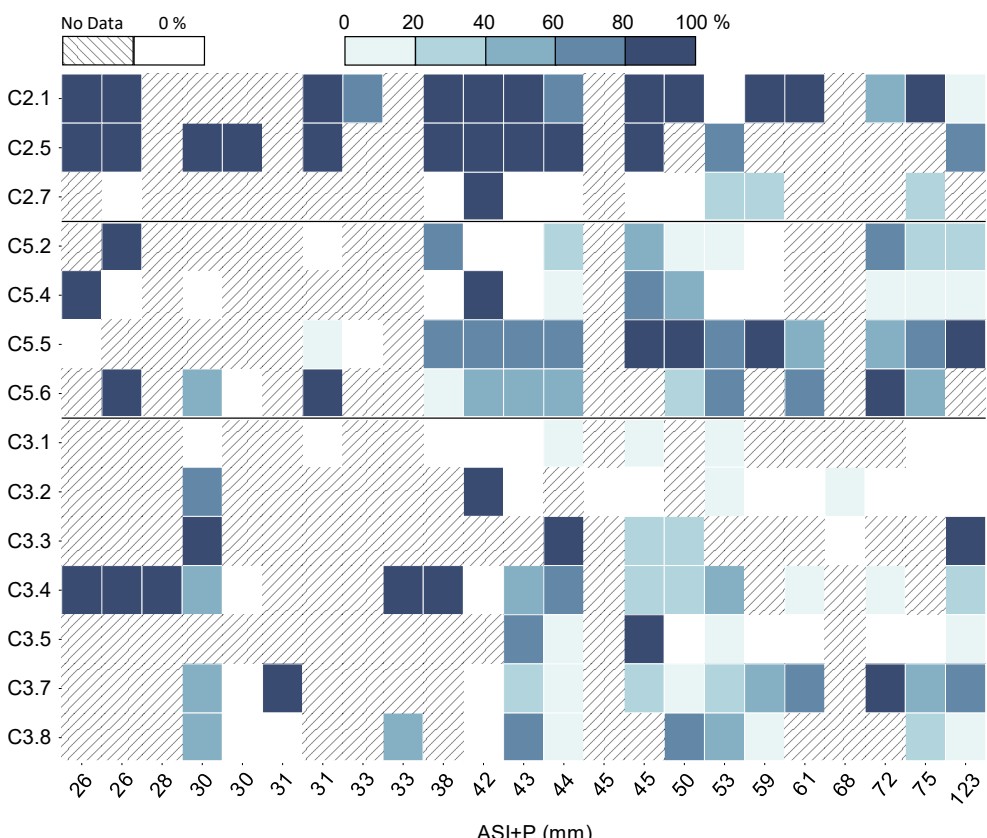

**Figure 5: The amount of OF as a fraction of total near-surface flow ($P_{OF}$) for each plot and each event (ordered by ASI+P). White cells indicate the lack of OF or TIF. Hatched cells indicate a lack of data for OF and TIF.**

### 4.4 Event responses and lag times

In Figure 6 we show the time series of OF and TIF for the event E20, a 20 mm rain event on the of August 30[th] 2022. All plots produced near-surface flow during this event except at the location C3.2, and the runoff ratios varied between 0.00 and 0.48 for OF and 0.01 and 0.91 for TIF. From this figure, it is clear that the amount of flow was largest for the plots with the highest TWI (as also indicated in Figures 3 and Table 3). The runoff response is fast for all plots (median time to rise ($t_r$): 0 min for OF and 5 min for TIF, ± 5 min), except for plot C3.5 for which TIF only started two hours after the start of the event. The responses to the two precipitation peaks during this event (at two hours and four hours after the start of the event) highlight the sensitivity (median time to peak ($t_p$): 13 min for OF and 15 min for TIF, ± 5 min) of the flow to changes in rainfall intensity. We observed two flow peaks for all plots that produced flow, especially for the plots for which the flow rate for TIF is higher than that for OF (e.g., C5.2 and C3.7).

The fast responses during this event are exemplary for all events. The time between the start of the rainfall event and the time to rise ($t_r$) was short over all events (median for all plot: 20 min for OF and 25 min for TIF, ± 5 min).





It was also short for the time between peak rainfall intensity and peak flow rate ($t_p$) (median for all events and plots: 15 min for OF and TIF, ± 5 min). The lag times were, on average, shortest for the wetland locations and longest in the forest and clearings (Figure S14). The Spearman rank correlation between vegetation cover and the response and peak lag times ($t_r$ and $t_p$) were statistically significant for TIF ($r_s$ = -0.6 and -0.5, respectively, p < 0.06 for both) but not for OF ($r_s$ = 0.2 and 0.2, p > 0.5 for both). Although these response times are clearly short,

they should be interpreted with caution as the precipitation was measured only at one location and the onset of precipitation probably varied across the catchment.

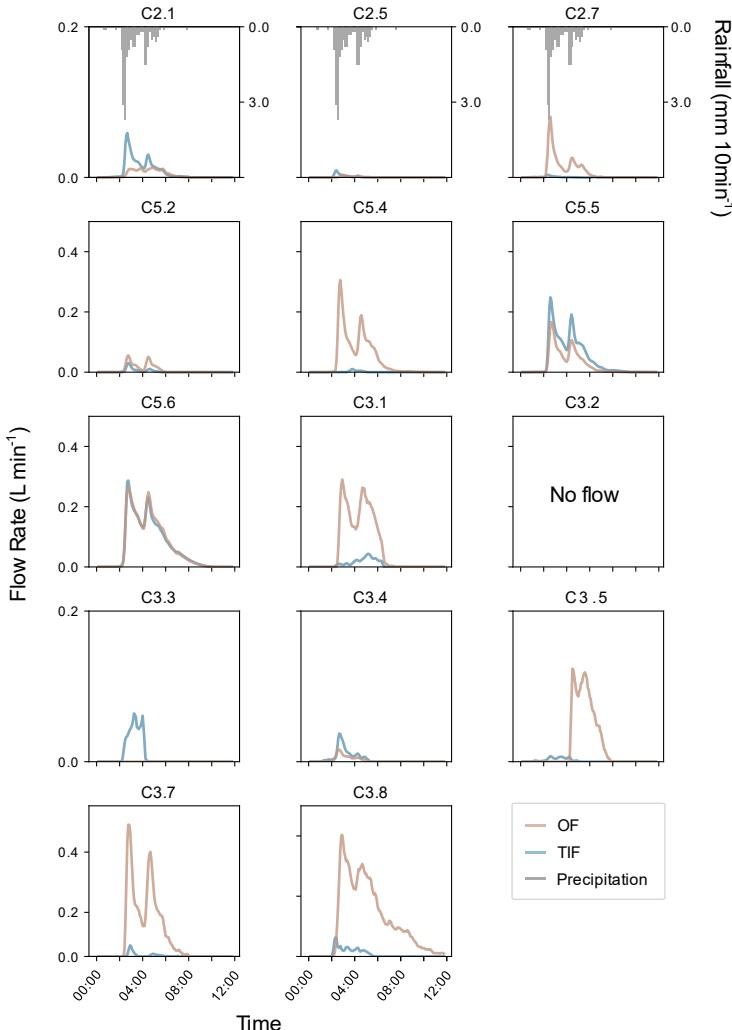

**Figure 6: Hydrographs for overland (OF; blue) and topsoil interflow (TIF; brown) for each plot during the 20 mm event on August 30th 2022 (event E20), as well as precipitation intensity (mm 10min⁻¹; only shown for the upper row of**

**figures). The same figure but with the y-axis extending to the range of observed flow rates for each subplot is shown in Figure S10.**



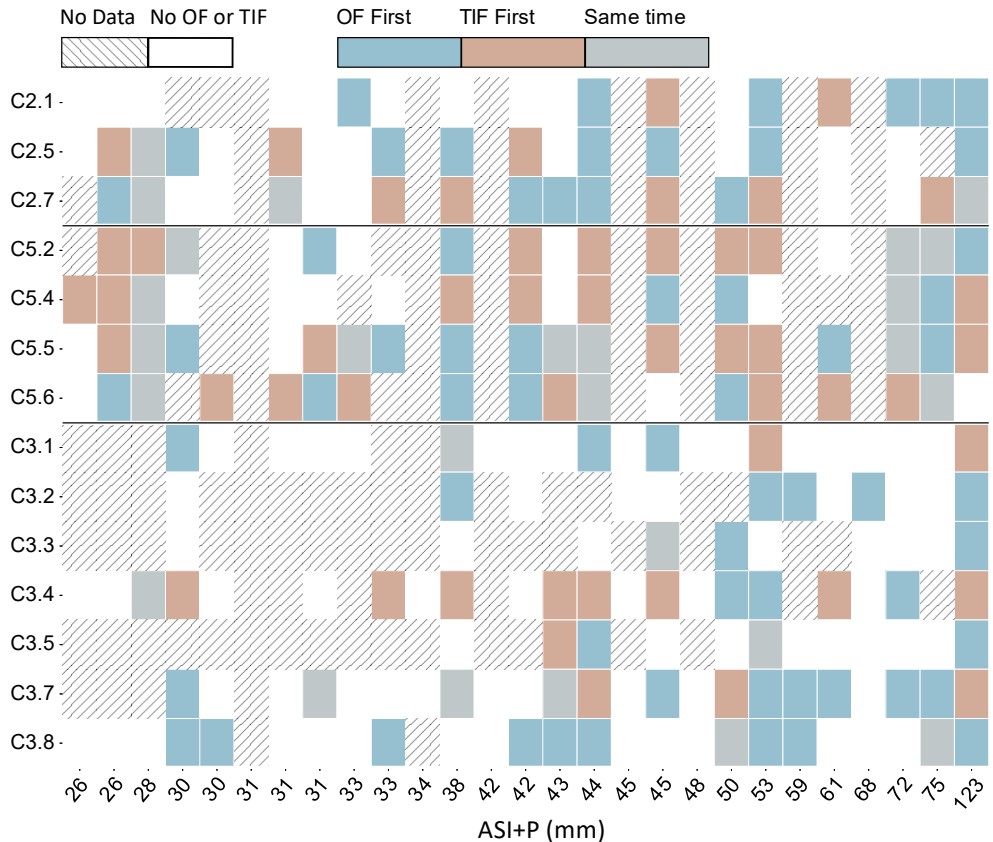

**Figure 7: Heatmap showing whether overland flow (OF, blue) or topsoil interflow (TIF, brown) responded first or if**
**both responded within 5 min (same time, gray) for each rainfall event (ordered by increasing *ASI+P*) and plot (y-axis).**
**Dashed lines indicate the lack of OF and TIF for that particular event, while white cells indicate a lack of data for either**
**OF or TIF. For a similar figure where the events are ranked by mean intensity, see Figure S11.**

For almost half (48%) of the cases (i.e., combinations of events and plots for which OF and TIF occurred) OF
responded first. For a third (34% of the cases) it was TIF, while for nearly a fifth of the cases (18%) OF and TIF
responded at the same time (i.e., within 5 min; Figure 6). OF responded more frequently first for the plots in
subcatchments C2 and C3 (48% and 61% of the cases) than for the plots in subcatchment C5, where OF occurred
at first only for 30% of the cases and TIF 46%. Whether OF or TIF responded first seemed unrelated to the event
characteristics though (Figure 7 and S11-13).

Peak flow occurred first for OF 41% of the cases, TIF also at 41%, and simultaneously for OF and TIF at 17%
(Figure S11) over the all catchment. However, there were differences between the subcatchments. For instance,
OF peaking first more often for the plots in C2 (62% of the cases), and TIF peaking more often first for the plots
in C5 (41% of the case; Figure S11).



## 5 Discussion

### 5.1 Near-surface flow occurs frequently

Near-surface flow was observed for many events, suggesting that it is a common runoff process in the Studibach catchment, even during a relatively dry summer. For half of the events, the $ASI+P$ was larger than the runoff generation threshold ($ASI+P \approx 39$ mm). A previous study using 50 cm-pipe-long overland flow collectors by Sauter (2017) during the summer and fall of 2016 suggested that OF also occurred frequently. To infer the frequency of occurrence for OF and TIF for periods beyond the summer measurement period, we looked at historical precipitation records. The estimated threshold for precipitation to generate OF and TIF was ~ 18 mm (range: 7-22 mm), which coincides with the threshold between 9-21 mm of Schneider et al. (2014) for a similar catchment in the Swiss pre-Alps. Using this threshold and the hourly precipitation data from the last 38 years for the snow-free season, we infer that considerable amounts of OF and TIF occur on average 28 events per year. When the $I_{mean}$ was $> \sim 2$ mm h$^{-1}$, more than half of the runoff plot generally started to produce OF and TIF (Figure S10). If we use this threshold to estimate the occurrence of near-surface flow together with hourly precipitation data, near-surface runoff occurred, on average across the catchment, for 23 events per year.

There are few studies to compare these frequencies of OF and TIF with. Still, measurements with overland flow collectors suggested that OF occurred for 10-90% of the events (depending on the location) in a forested catchment in Panama (Zimmermann et al., 2014) and for 44% (range: 0-71%) of the events in agricultural fields (Vigiak et al., 2006). Measurements at runoff plots suggested that OF occurred for 55% of the events in fallows in Madagascar (Zwartendijk et al., 2020). Biomat flow occurred for 50% of the events in moso-bamboo forested sites in Japan (Ide et al., 2010). Thus, although these sites are all very different, the occurrence of near-surface runoff for almost half of the event seems to be a reasonable approximation.

### 5.2 Occurrence of near-surface flow varies spatially

The frequency of near-surface runoff varied spatially and ranged between 14-78% for OF and 19-86% for TIF. This variation was mainly linked to vegetation cover ($r_s = 0.35$ for OF and 0.64 for TIF) and TWI ($r_s = 0.51$ for OF and 0.51 for TIF). In the Studibach, these two variables are related to each other ($r_s = 0.60$ for the 14 plots; Table S3) as the steeper locations near the ridges with a low TWI are mainly covered by forests and the wetter flatter areas with a high TWI are mostly wetlands. The frequency of near-surface flow being related to TWI is not surprising as Rinderer et al. (2014) already demonstrated that less rain is needed for the groundwater levels to start rising for sites with a higher TWI. Indeed, for the wetlands, where the TWI was highest, the occurrence of OF and TIF was highest (> 70%) and the lag times for OF and TIF were shortest (median $t_r$: 17min; $t_p$: 15min). Thus, vegetation and TWI are good indicators for spatial variation in the frequency of near-surface flow.

However, there were also exceptions to the relation between the frequency of OF and TIF and vegetation or TWI. For the forested plots C2.1 and C2.5, OF was measured much more frequently than TIF and more frequently than expected based on their TWI. These plots are covered by a thick moss layer (see photos in Figure S2 and S3). It appears that the boundary between the biomat with the moss and shallow subsurface may create an interface of low infiltration or hydrophobicity (cf. Gall et al., 2024; Gerke et al., 2015), especially when $ASI$ was low, that could have promoted the occurrence of biomat flow, which was included in the OF. According to Pan et al. (2006),





moss cover should reduce surface runoff by absorption and retention. As we observed frequently OF for these plots, it suggests that on steep slopes, a thick moss layer could induce biomat flow. This inference is supported by the observation that the $R_{OF}$ for these plots remains similar with increasing wetness conditions ($ASI$; Figure 4). In the lower Studibach (e.g., plots C3.1 and C3.5), the forested plots were covered by a scattered moss layer and

grass, as well as some forest litter (needles and leaves), which likely reduced surface runoff and increased infiltration. The rooting system can create fast infiltration and lateral subsurface pipe flow, prompting TIF instead of saturated conditions and OF (see also section 5.3).

For two grassland locations, C5.2 and C5.3, the occurrence of TIF was high, but the occurrence of OF was relatively low, even though the sites have a relatively steep slope (mean = 34°). These two locations were subject

to cattle trampling, suggesting they may quickly become saturated (Monger et al., 2022; Wheeler et al., 2002). Although we expected this to lead to more OF, they did not generate as much OF as expected (mean $R_{OF}$ over the events of 0.04 and 0.06, respectively) compared to TIF (mean $R_{TIF}$ over the events of 0.16 and 0.16), suggesting most flow occurred through the topsoil and OF was generated locally. Thus, instead, it appears that the presence of holes from trampling could lead to ponding of water on the surface (Pietola et al., 2005), which promotes

infiltration and increases the roughness for OF.

### 5.3 Threshold runoff response

The Antecedent Soil-moisture Index and precipitation depth ($ASI + P$) thresholds (calculated for the top 5 cm of soil) ranged from 29-55 mm for OF and from 17-70 mm for TIF. They were generally lowest for the wetter locations (Figure 4). The main factor influencing the runoff threshold was the TWI, a good indicator of the wetness

conditions (Beven and Kirkby, 1979) and groundwater levels (Rinderer, van Meerveld, et al., 2014). Indeed, the Spearman rank analysis indicates that $R_{OF}$ was negatively correlated to $ASI$ for plots with a low TWI and positively correlated for plots with a high TWI (Figure S5).

For TIF, the threshold increased with the slope gradient ($r_s = 0.80$; p < 0.01), which is reflected in the inverse relation between $R_{TIF}$ and slope as well (Table 3). For OF the relation with slope was less clear, as for some of the

steeper plots (e.g., C2.1 and C2.5) we could not define a clear a threshold (Figure 4). Generally, OF rates increase with slope (Essig et al., 2009; Haggard and Moore, 2005; Morbidelli et al., 2013) but do not have to do so in a linear continuous way (Jourgholami et al., 2021; Komatsu et al., 2018). Interestingly, for the plots with a slope higher than 22°, TIF thresholds became higher than OF thresholds, suggesting that more rain is required to generate TIF than OF above 22°, which follows findings saying that infiltration time is less on steep slopes and thus inducing

some OF (Battany and Grismer, 2000; Mumford and Neal, 1938). Nevertheless, the runoff ratios for the steeper slopes were smaller (mean $R_{OF}$: 0.03) than for the other plots (mean $R_{OF}$: 0.34; see Figures 3 and 4), probably due to the lack of return flow from outside the plots (see section 5.4).

### 5.4 Inference of runoff mechanisms

We did not observe a relation between near-surface runoff and the maximum precipitation intensity (Table 2).

Instead, OF could be explained by the $ASI+P$ threshold. This suggests that OF is saturated overland flow and not Hortonian (i.e., infiltration excess) overland flow. The runoff ratios > 1 for OF suggest that OF consists for at least some part of the return flow from outside the plot. Return flow is likely more important for the flatter sites with wetland and grass vegetation, which explains the inverse relation between slope and OF ratios. Only for the



forested sites with a thick moss cover we did frequently observe OF but no TIF. Biomat flow is likely an important runoff mechanism as well, e.g. flow through the moss layers in C2.1 and C2.5 (see section 5.1), but also at other plots. Biomat flow can explain the earlier onset of OF than TIF for half of all events and plots (Figure 7). However, both OF and TIF responded relatively quickly to rainfall (Figure S14) and changes in the rainfall intensity (Figure 6).

The $ASI+P$ threshold was very similar for OF and TIF. Together with the fact that at most plots OF and TIF both occurred and that both flow pathways responded quickly to changes in precipitation intensity, this suggests that the processes generating OF and TIF are highly related. That the runoff ratios for OF did not change considerably after we removed the border at the top of the plots (Figure 4) suggests that OF flow pathways on the surface are rather short. Thus, there is likely considerable interaction between OF and TIF and that OF water infiltrates into the surface after a short distance, while at other places TIF exfiltrates. However, this requires further research using tracers.

The runoff ratios for OF and TIF (over all plots and events with range between plots: median: 0.8% (0-5%), and mean: 17% (0-147%) for OF; median: 2% (0-10%), and mean: 24% (0-87%) for TIF) are in the range of those for rainfall simulation studies in the Swiss pre-Alps (1-22% for $R_{OF}$; Schneider et al., 2014) and the Austrian Alps (0-85% for $R_{OF}$; Meißl et al., 2023). Some of the large responses during medium events (e.g., 0 to 77% (median: 4%; mean: 13%) for OF and 0-79% (median: 7%; mean: 16%) for the 20 mm event on 30 August 2022; Figure 6), suggest that these processes can be important for stormflow generation. This is one of the few studies worldwide that collected field data of OF and TIF for a densely vegetated catchment to study their spatiotemporal variability. These findings may contribute to development and testing of models to estimate the relative importance of OF and TIF as well as catchment scale hydrological models for the region to ensure that they simulate the quick response to precipitation for the right reasons. However, further studies are needed to determine the connectivity of these near surface flow pathways to the stream network and their importance a catchment scale.

## 6 Conclusions

Overland flow (OF) and topsoil interflow (TIF) were measured for 14 small plots across a small pre-Alpine catchment during the summer and fall of 2022. OF and TIF occurred frequently almost at all plots. For most plots, runoffs occurred after antecedent soil moisture ($ASI$ over the top 5 cm of the soil) and precipitation ($P$) exceeded 39 mm or a precipitation threshold of ~ 18 mm had been reached. These conditions occur frequently and suggest that OF and TIF also occur frequently. However, there was considerable spatial variation across the catchment. The frequency of OF and TIF occurrence and the runoff ratios for OF and TIF were correlated to the topographic wetness index (TWI) and vegetation cover. Wetter sites (grassland and wetland) produced more flow, and more often. For the plots in the forest and natural clearings in the forest, the occurrence of OF and TIF was more variable, but overall, they produced less runoff and less often. However, there were some exceptions. For some forested plots (e.g., C2.1 and C2.5), OF occurred frequently and OF rates were higher than for TIF. For these plots, biomat flow at or through the moss layer was likely important. The high runoff ratios for OF for some sites (> 1) highlight the importance of return flow. The runoff ratios for OF were not affected by the opening of the plot borders, suggesting that OF pathways are short. The fast response of both flow pathways highlights the importance of preferential flow and suggests considerable interaction between OF and TIF. Although these plot scale studies highlight the frequent occurrence of near-surface runoff processes across the entire catchment, their importance



for stormflow generation at the catchment scale depends on their connectivity to the stream network and thus requires further research.

**Data availability**

Data can be provided by the corresponding authors upon request and will be uploaded on the repository EnviDat.ch.

**Author contributions**

VAG and IvM conceptualized the study. VAG, AL, IvM organized the data collection. VAG and AL performed

the data collection; VAG analyzed the data; VAG wrote the manuscript draft; VAG, AL and IvM reviewed and edited the manuscript. IvM supervised the project.

**Competing interests**

The authors declare that they have no conflict of interest.

**Acknowledgements**

We thank our colleagues from the Swiss Federal Institute for Forest, Snow and Landscape Research (WSL), Mountain Hydrology Group for collaborating in the Alptal and sharing the data of Erlenhöhe Meteorological Station. We thank our colleagues from the H2K department at the University of Zurich for their help in the field. We thank Manfred Stähli and Jan Seibert for their advice and comments on this manuscript. We thank Oberallmeindkorporation Schwyz (OAK), the Department of Environment of the Canton of Schwyz and the

municipality Alpthal for their cooperation. This research was conducted as part of the TopFlow: (in)visible water flows near the surface project funded by the Swiss National Science Foundation (Grant 197194).

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
