# Peer review of "When and where does near-surface runoff occur in a pre-Alpine headwater catchment?"

_Hydrology and Earth System Sciences, 2024_

## Author Comment (AC1)

Dear Anonymous Referee #1

We would like to thank you for reviewing our manuscript, for your comments and suggestions, and for pointing out some minor mistakes. These comments help us to improve the manuscript. We respond to the individual comments in blue font below.

I believe that this study will be interesting to many readers. The research was very well designed. It documented considerable variability in runoff response. The manuscript provides clear and detailed information for the readers to understand the results and make their own interpretation/conclusions.

Thank you for these kind words and for valuing our study.

Reference to climate change in lines 72-73 is not needed, the topic is interesting in itself.

Indeed, climate changes is not the topic of the manuscript. We wanted to point that changes in rainfall patterns may lead to changes in the frequency of overland flow and the importance of near surface flow pathways for runoff generation but we will remove these sentences.

l. 270 - I did not find figure S4 in the Supplement

Thank you for spotting this. We referred to Figures S4 and S5 but should have referred to Figures S8 and S9. We will fix this in the revised version of the manuscript and double check all references to figure numbers. In addition, we will change the supplement so that it first shows all the tables and then all the figures. This will make it easier to find a specific figure.

l. 310 - mentions relations with TWI and local slope. It might be good to note that TWI considers slope as well.

That is correct and we will add this explanation. However, note that the TWI was calculated by Rinderer et al. (2014) based on a smoothed DEM, while the slope mentioned here is the local slope for the plot measured in the field. These two are not the same because they are based on different data.

Fig. 5 shows that that there was quite a lot of "No data" for events with ASI+P smaller than approximately 38 mm. Could that have an influence on the interpretation of results?

Indeed, there is a lot of no data for the small events (with ASI+P smaller than 38 mm) in this figure. This is partly due to the lack of data due to sensor failures for some plots for some events but the main reason is that for these smaller events there was often no flow for OF and for TIF (see Figure 3). As a result, the importance of OF for total near surface flow could not be determined (i.e., the fraction would be 0/0). This is indicated in Figure 5 as no data. We will update the legend of Figure 5 to highlight that the hashed lines indicate no flow or no data. Alternatively, we can try to use different hashed lines for no data and for no flow. If there was OF but no TIF, it is shown as 100% OF, and vice versa as 0% OF.
We will explicitly point out that we only have data for a small number of plots for the small events in the text, that these are mainly the forested plots with moss, and that

this may have influenced the correlations somewhat. However, overall, the lack of data for these types of events mainly reflects the threshold response for most plots.

l. 337 - Is it possible to say why was namely event on August 30 chosen? Fig. S8 shows that there two events with total precipitation of 20 mm with enough data recorded. Is it possible/useful to comment on similarities/differences (and their probable reasons) of runoff ratios at the same plots during those two events?

This event was chosen for several reasons: a) we had data and measured flow for most plots, b) it is a medium sized event that is quite common for the Studibach, and c) there were two clear rainfall peaks during the event, which are interesting to observe. We made a similar figure for event E22 on the 14th of September 2022. This is a 20 mm event with relatively wet antecedent conditions (ASI+P = 42 mm). We will add this figure to the supplementary materials and refer to it in the text.

[Figure]

**Figure S10: Hydrographs for overland (OF; blue) and topsoil interflow (TIF; brown) for each plot during the 20 mm event on 14th September 2022 (event E22), as well as precipitation intensity (mm per 10 min;**

only shown for the upper row of figures). The plot name is indicated above each subplot. All loggers recorded data during the event, but for many plots there was either no flow or a very small flow rate, which appears as a horizontal line at zero.

For the consistency with the main text it would be better to write figure captions in the Supplement below the figures as well.

We will do this and also make sure that all the tables have the same format. In addition, we will move all the tables to the beginning of the document and the figures to the end, so that it is easier to find a specific figure.

---

## Author Comment (AC2)

**Dear Anonymous Referee #2**

We would like to thank you for reading our manuscript and the helpful comments and suggestions, and for pointing out some unclarities. These comments will help us to improve the manuscript. We respond to the individual comments in blue font below.

This study used hydrological measurement network consisting of 14 small 60 runoff plots (1 m x 3 m) across the 20 ha Studibach catchment in the Alptal, Switzerland to analyze the occurrence of OF and TIF, their controlling factors and threshold. One of the major concerns is that at plot-scale, soil properties mainly governs the runoff dynamics, however, this study didn't discuss role of soil characteristics (texture, hydraulic conductivity and parameters of soil water retention curve) in explaining variation of OF and TIF.

Thank you for this remark. The entire catchment is underlain by gleysols and flysch bedrock. Therefore, we don't expect large scale or systematic differences in the soil properties due to differences in geology.

One of the main reasons that we didn't focus much on soil properties is that our study aimed to see how site characteristics that can easily be determined across a catchment, influence OF and TIF. Slope and vegetation are two of the main factors affecting the soil properties (considering the similarity in geology, climate and age across the catchment). Therefore, we would expect a large part of the potential effect of differences in soil properties to be included in the relations with slope, TWI or vegetation. Previous studies have shown that groundwater levels and vegetation in the catchment are both highly related to the topographic position (e.g., Rinderer et al., 2014), and therefore we would expect soil properties to also be (at least partly) related to topographic position and vegetation.

Nevertheless, we have determined the soil properties based on soil cores (Hyprop method), and the saturated hydraulic conductivity (Ksat) of the surface soil based on double ring infiltrometer measurements. However, the number of measurements is small (1 per plot) and the variability is relatively large (in part due to the high organic carbon content for some cores). We think that a large part of the observed variation in soil properties would also be observed if we had taken multiple samples per plot (but don't have the data to show this). We don't see any systematic variation in the soil properties with vegetation (Tables 1 and 2) and only the carbon content (based on the loss on ignition) and the porosity (based for a soil core) are weakly related to TWI (Table 2). Note that the carbon content and porosity were highly correlated themselves ( $r_s=0.95$ ;  $p=8.2 \times 10^{-7}$ ).

We, nevertheless, calculated the Spearman rank correlation ( $r_s$ ) between the runoff ratios and the soil properties. Table 3 shows for how many events these correlations were statistically significant (p<0.1). Overall, the correlations are very low, significant for only a few events, and that the direct relationship is often not consistent from event to event. The exception is the relation between clay content at 10-15 cm and the runoff ratio for OF which was significant for 11 of the 27 events. This can largely be explained by the low clay content but high silt content for two samples (Figure 1), one from a plot located in the forest and one from a plot in a wetland. Both plots have a high moss cover and frequently produced OF, leading to the negative relation between the runoff ratio for OF and clay content for some events, particularly events that did not produce a lot of OF at the majority of plots. We think that this result is a spurious correlation because the two samples have a low clay content but high silt content. It is well possible that the silt content in these samples is higher than expected based on the other samples because they did not completely disaggregate (leading to a measurement error and overestimation of the silt content), or because they may fall on different sides of the boundary between silt and clay. The two plots did not have an abnormal clay content at 2-7 cm and had, as explained in the paper, high OF ratios for different reasons. For the forested plot, OF occurred through or below the moss as biomat flow due to hydrophobicity of the underlying soil when it was dry and in the wetland OF occurred frequently due to the low storage capacity and frequent saturation, as well as return flow.

We will add some discussion on the correlations with the soil properties to the revised manuscript and add a table (similar to Tables 1 and 3 in this response), to the supplementary material.

Table 1: Average values and range (min-max) for the soil properties for forested and grassland plots, as well as the number of measurements für each property (n). Note that we grouped the plots in the forest and clearings and the plots in the grasslands and wetlands for this analysis because of the small number of samples (i.e., low n).

|                                   | Depth         | Forest | and Clearin | ıg | Grassland and |           |   |  |
|-----------------------------------|---------------|--------|-------------|----|---------------|-----------|---|--|
|                                   | ( cm ) |        |             |    | wetlands      |           |   |  |
|                                   |               | Mean   | Range       | n  | Mean          | Range     | n |  |
| Porosity (%)                      | 2-7           | 83     | 78-94       | 9  | 84            | 73-92     | 3 |  |
|                                   | 10-15         | 78     | 67-94       | 8  | 75            | 52-91     | 4 |  |
| Water content at field            | 2-7           | 54     | 33-75       | 9  | 69            | 65-73     | 3 |  |
| capacity (%)                      | 10-15         | 51     | 41-78       | 8  | 64            | 49-73     | 4 |  |
| Water content at                  | 2-7           | 30     | 20-43       | 9  | 28            | 16-37     | 3 |  |
| wilting point (%)                 | 10-15         | 31     | 15-52       | 8  | 30            | 20-34     | 4 |  |
| Drainable porosity                | 2-7           | 22     | 8-42        | 9  | 11            | 8-17      | 3 |  |
| (%)                               | 10-15         | 19     | 4-35        | 8  | 24            | 1-68      | 4 |  |
| Carbon content (%)                | 2-7           | 28     | 19-69       | 8  | 43            | 32-54     | 2 |  |
|                                   | 10-15         | 22     | 11-48       | 8  | 21            | 3-43      | 4 |  |
| Sand content (%)                  | 2-7           | 11     | 6-22        | 7  | 25            | -         | 1 |  |
|                                   | 10-15         | 14     | 6-28        | 6  | 20            | 18-21     | 2 |  |
| Silt content (%)                  | 2-7           | 38     | 31-44       | 7  | 40            | -         | 1 |  |
|                                   | 10-15         | 35     | 32-55       | 6  | 53            | 41-66     | 2 |  |
| Clay content (%)                  | 2-7           | 52     | 47-57       | 7  | 35            | -         | 1 |  |
|                                   | 10-15         | 52     | 18-58       | 6  | 27            | 15-39     | 2 |  |
| Bulk density (g/cm 3 ) | 2-7           | 0.5    | 0.15-0.57   | 9  | 0.52          | 0.21-0.78 | 4 |  |
|                                   | 10-15         | 0.6    | 0.16-0.86   | 8  | 0.57          | 0.23-1.27 | 5 |  |
| K sat (mm/h)           |               | 122    | 8-320       | 6  | 345           | -         | 1 |  |
| Depth A (cm)                      |               | 17     | 10-20       | 9  | 12            | 5-15      | 5 |  |
| Depth B (cm)                      |               | 35     | 30-40       | 9  | 37.2          | 31-42     | 5 |  |

Table 2: Spearman rank correlation  $(r_s)$  between the soil properties measured at 10-15 cm depth and the topographic wetness index (TWI) and vegetation (ordered as in the manuscript: 0: forest, 1: clearing, 2: grassland, 3: wetland). Statistically significant correlations (p<0.1) are indicated in bold font. The number of data points used for the correlation is given in parentheses after the soil properties.

|                                      | Т     | WI      | Vegetation |         |  |  |
|--------------------------------------|-------|---------|------------|---------|--|--|
|                                      | rs    | p-value | rs         | p-value |  |  |
| Porosity (13)                        | 0.55  | 0.05    | 0.30       | 0.32    |  |  |
| Water content at field capacity (13) | 0.41  | 0.17    | 0.16       | 0.60    |  |  |
| Water content at wilting point (13)  | -0.25 | 0.42    | -0.31      | 0.30    |  |  |
| Drainable porosity (13)              | -0.21 | 0.50    | 0.02       | 0.94    |  |  |
| Carbon content (13)                  | 0.56  | 0.05    | 0.30       | 0.33    |  |  |
| Sand content (9)                     | 0.21  | 0.61    | 0.24       | 0.57    |  |  |
| Silt content (9)                     | -0.12 | 0.78    | 0.43       | 0.29    |  |  |
| Clay content (9)                     | -0.55 | 0.16    | -0.50      | 0.20    |  |  |
| $K_{sat}$ (7)                        | -0.14 | 0.76    | 0.42       | 0.35    |  |  |
| Depth A (14)                         | 0.42  | 0.14    | 0.02       | 0.96    |  |  |
| Depth B (14)                         | 0.22  | 0.45    | 0.14       | 0.63    |  |  |

Table 3: Number of events (out of 27) for which there was a significant positive or negative significant (p<0.1) Spearman rank correlation between the soil properties measured for a core taken at either 2-7 or 10-15 cm below the soil surface and the runoff ratio for OF or TIF.

| Flow pathway                        | OF            |   |          | TIF |               |   |          |   |
|-------------------------------------|---------------|---|----------|-----|---------------|---|----------|---|
| Depth                               | 2-7 cm |   | 10-15 cm |     | 2-7 cm |   | 10-15 cm |   |
| Positive (+) or negative (-) | +             | - | +        | -   | +             | - | +        | 1 |
| correlation                         |               |   |          |     |               |   |          |   |
| Porosity                            | 2             | 0 | 6        |     | 2             | 0 | 3        | 0 |
| Water content at field capacity     | 1             | 0 | 2        | 1   | 0             | 0 | 0        | 2 |
| Water content at wilting point      | 0             | 4 |          | 2   | 0             | 1 | 0        | 3 |
| Drainable porosity                  | 0             | 1 | 0        | 0   | 0             | 1 | 0        | 0 |
| Carbon content                      | 3             | 2 | 5        | 0   | 1             | 1 | 4        | 0 |
| Sand content                        | 4             | 0 | 6        | 0   | 1             | 1 | 0        | 4 |
| Silt content                        | 3             | 0 | 6        | 0   | 0             | 0 | 2        | 0 |
| Clay content                        | 0             | 4 | 0        | 11  | 1             | 0 | 0        | 0 |
| K sat                    | 0             | 0 | 0        | 0   | 0             | 0 | 0        | 0 |
| Depth A                             | 3 (+); 1(-)   |   |          |     | 3(+); 1 (-)   |   |          |   |
| Depth B                             | 0             |   |          |     | 2 (+)         |   |          |   |

Figure 1. Texture triangle for soil samples taken from the plots.

Further, It is not very clear how TIF was measured in the field. Please explain it.

Thank you for pointing this out. We described this on lines 133-136 but now realize that we accidentally may have confused the reviewer by referring to **lateral subsurface flow** here. The lateral subsurface flow through the topsoil is **TIF**. We will make this clearer in the revised manuscript:

"At the lower end of the plot, we dug a trench until the depth of the reduced clay layer (generally at ~ 40 cm below the soil surface; Table 1), where there are only very few visible roots. We put drain foil on the trench face to block the **lateral subsurface flow through the topsoil** and a drainage tube at the bottom of the trench (rolled into the foil) to collect this **TIF** and channel it via a hose to an Upwelling Bernoulli Tube."

Additionally, we will add the following figure to the supplementary material.

Figure 2. Schematic representation of the collection system at the bottom of each plot with the gutter to collect overland flow and biomat flow, and the drainage mat and drainage tube in the trench to collect TIF, and the routing of the water to the Upwelling Bernoulli Tubes

**Line 194: On what basis authors divided the low-, medium-, and high- intensity rainfall ranges.**

Thank you for your comment. We realize that this was not well explained and admit that this is a bit arbitrary. We looked at the distribution of the mean intensities for the different events (see Figure 3) and selected three different ranges so that each intensity class would include several events. These intensity classes are similar to the precipitation intensity ranges for 12 hour events given by Meteoswiss for different danger classes for the northern part of the Alps: 1.7 mm h-1 (danger level 2), 2.9 mm h-1 (danger level 3), and 5 mm h-1 (danger level 4), but are slightly different in order to ensure a more even spread of the events among the three classes. We will make this clearer in the revised version of the manuscript.

---

## Author Response (AR1)

Dear Editor,

Thank you for your comments and handling the review process of our manuscript.
Please find below the changes that we made to the manuscript in response to the reviewers' comments. We refer to the responses to the reviewers' comments for more extensive answers to the review comments. In short, we have made all the changes that we promised to make during the discussion phase of the review process. In addition, we fixed some unclarities in the text and minor mistakes that we found in the previous version of the manuscript. None of these additional changes affect the conclusions of the work.
* * *
reviewer #1 comments
* * *
Reference to climate change in lines 72-73 is not needed, the topic is interesting in itself.

> We have removed these sentences.

l. 270 - I did not find figure S4 in the Supplement

> We have fixed this in the revised version of the manuscript and double checked all references to figure numbers. In addition, we have changed the supplementary materials so that it first shows all the tables and then all the figures. This will make it easier to find a specific figure.

l. 310 - mentions relations with TWI and local slope. It might be good to note that TWI considers slope as well.

> We agree and have added explanation to the revised manuscript (l.310).

Fig. 5 shows that that there was quite a lot of "No data" for events with ASI+P smaller than approximately 38 mm. Could that have an influence on the interpretation of results?

> We now explicitly point out that we only have data for a small number of plots for the small events (l.327), that these are mainly the forested plots with moss cover, and that this may have influenced the correlations somewhat.

l. 337 - Is it possible to say why was namely event on August 30 chosen? Fig. S8 shows that there two events with total precipitation of 20 mm with enough data recorded. Is it possible/useful to comment on similarities/differences (and their probable reasons) of runoff ratios at the same plots during those two events?

> It was just an example event for which we recorded flow for a large number of plots. We have added a figure for another similar event to the supplementary materials and refer to it in the text (l.345).

For the consistency with the main text it would be better to write figure captions in the Supplement below the figures as well.

> We did this and also made sure that all the tables have the same format. In addition, we have moved all the tables to the beginning of the document and the figures to the end, so that it is easier to find a specific figure.

This study used hydrological measurement network consisting of 14 small 60 runoff plots (1 m x 3 m) across the 20 ha Studibach catchment in the Alptal, Switzerland to analyze the occurrence of OF and TIF, their controlling factors and threshold. One of the major concerns is that at plot-scale, soil properties mainly governs the runoff dynamics, however, this study didn't discuss role of soil characteristics (texture, hydraulic conductivity and parameters of soil water retention curve) in explaining variation of OF and TIF.

> We have added some discussion (l.313) on the correlations with the soil properties to the revised manuscript and additional tables to the supplementary material.

Further, It is not very clear how TIF was measured in the field. Please explain it.

> We have updated the manuscript (l.133) to point out that it was TIF (subsurface flow through the topsoil) that was collected. Additionally, we have added a schematic figure to the supplementary material that shows the monitoring setup.

Line 194: On what basis authors divided the low-, medium-, and high- intensity rainfall ranges.

> We have added some explications in the manuscript (l.197-201) to make this clearer in the revised version of the manuscript and refer to the Meteoswiss classification.

Topsoil interflow (TIF): at what depth TIF takes place?

> We have added a better description on how we collected TIF and highlight the depth of TIF occurrence more clearly as well (l.135). Additionally, we have added a figure in the supplementary material.

Line 435: "Indeed, the Spearman rank analysis indicates that ROF was negatively correlated to ASI for plots with a low TWI and positively correlated for plots with a high TWI (Figure S5)": what would be the possible reason?

> We have added details about the mechanisms in the manuscript (l.443) to make this clearer.

Line 490: The fast response of both flow pathways highlights the importance of preferential flow and suggests considerable interaction between OF and TIF. How the fast response of OF highlights the importance of preferential flow.

> We have added information (l.471) in the revised manuscript and give a few more details.

Add legend titles to Figures 2 and 3.

> We have added the legend titles in the new version of the manuscript.

Add legend titles to Figures S8 and S9.

> We have added the legend titles in the new version of the manuscript.